# Two Biosensors for the Determination of Interleukin-6 in Blood Plasma by Array SPRi

**DOI:** 10.3390/bios12060412

**Published:** 2022-06-14

**Authors:** Beata Szymanska, Zenon Lukaszewski, Lukasz Oldak, Beata Zelazowska-Rutkowska, Kinga Hermanowicz-Szamatowicz, Ewa Gorodkiewicz

**Affiliations:** 1Bioanalysis Laboratory, Faculty of Chemistry, University of Bialystok, Ciolkowskiego 1K, 15-245 Bialystok, Poland; b.szymanska@uwb.edu.pl (B.S.); l.oldak@uwb.edu.pl (L.O.); 2Faculty of Chemical Technology, Poznan University of Technology, pl. Sklodowskiej-Curie 5, 60-965 Poznan, Poland; 3Doctoral School of Exact and Natural Science, Faculty of Chemistry, University of Bialystok, Ciolkowskiego 1K, 15-245 Bialystok, Poland; 4Department of Pediatric Laboratory Diagnostics, Medical University of Bialystok, Waszyngtona 17, 15-274 Bialystok, Poland; zelazowskab@wp.pl; 5Department of Clinical Oncology, Comprehensive Cancer Center, Ogrodowa 12, 15-027 Bialystok, Poland; kiniaziel@wp.pl

**Keywords:** interleukin-6, array SPRi, cancer biomarkers, ovarian cancer, blood plasma

## Abstract

Interleukin-6 (IL-6) is a biomarker of inflammation, the advanced stage of COVID-19, and several cancers, including ovarian cancer. Two biosensors for the determination of IL-6 in blood plasma by array SPRi have been developed. One of these biosensors consists of the mouse monoclonal anti-IL-6 antibody as the receptor immobilized via the cysteamine linker. The second contains galiellalactone as the receptor, being an inhibitor specific for IL-6, immobilized via octadecanethiol (ODM) as the linker. Both biosensors are specific for IL-6. The biosensor with the antibody as the receptor gives a linear analytical response between 3 (LOQ) and 20 pg mL^−1^ and has a precision between 8% and 9.8% and recovery between 97% and 107%, depending on the IL-6 concentration. The biosensor with galiellalactone as the receptor gives a linear analytical response between 1.1 (LOQ) and 20 pg mL^−1^, and has a precision between 3.5% and 9.3% and recovery between 101% and 105%, depending on IL-6 concentration. Both biosensors were validated. Changes in IL-6 concentration in blood plasma before and after resection of ovarian tumor and endometrial cyst, as determined by the two developed biosensors, are given as an example of a real clinical application.

## 1. Introduction

Body fluids such as blood plasma or serum, urine, saliva, and cerebrospinal fluid, etc., contain a vast amount of potentially useful diagnostic information. This information is so far unavailable due to the lack of methods for its discovery. Several dozen currently used molecular biomarkers, such as Troponin T and I (cardiovascular disease), PSA, CA 125, HE 4, and CEA (cancer), show the usefulness of the diagnostic information contained in body fluids and the potential of the so-called ‘liquid biopsy’. The discovery of new biomarkers, along with progress in their determination, can be expected to revolutionize diagnostics. New tools need to be developed to screen the population for early diagnosis of cancer or neurodegenerative diseases. Moreover, the biomarkers currently used do not provide entirely certain information: none of them ensures a 100% correct diagnosis, that is, a complete absence of false positive results and of false negative results. To improve the correctness of diagnosis, two or more biomarkers are determined. However, oncologists’ expectations are much higher: the level of biomarkers should supply information about the stage of the disease and the effectiveness of therapy.

Biosensors are one of the types of tools for extracting information about biomarkers in body fluids. Various types of biosensors are used based on techniques such as electrochemical methods, immunofluorescence, colorimetry, surface-enhanced Raman scattering (SERS), nuclear magnetic resonance, and surface plasmon resonance (SPR) [1]. The SPR measuring techniques exploit the surface plasmon effect, occurring in a nanometric layer of metal on glass. Incident light interacts with plasmons in the metal, which causes a lowering of reflectivity. The effect occurs at a certain angle of incident light and is known as the SPR dip. The SPR effect is observed on gold, silver, copper and aluminum. Covering the metal surface with organic substances results in a shift of the angle of reflected light. When polarized incident light is used, changes in the angle of polarization are observed. There is also a red shift in the absorption spectrum. All of these effects are used in analytical applications of the SPR effect based on different measuring techniques.

Fluidic SPR is the most popular SPR technique in analytical applications, e.g., [2,3,4,5]. The Kretschamann configuration of an SPR instrument is usually applied. A biosensor is formed on the prism covered with a nanometric-size metal (usually gold) or, more frequently, on a glass slide covered with the nanometric-size metal (usually gold) fixed on the prism with an immersion oil. The analyzed sample flows through a measuring cell, pushed by a buffer. The SPR signal is measured continuously. A plot of the SPR signal against time is called a sensorgram. A biosensor is formed in situ during measurement. Finally, a cleaning solution is directed into the cell for regeneration of the biosensor. Usually, two parallel channels are used; one of them serving as a reference channel. The highest value on the sensorgram is taken to be the analytical signal. The conversion of an SPR signal to an image using a CCD camera is called the SPR Imaging (SPRi). More information about recent advances in Kretschmann configuration for SPR sensors based on two-dimensional materials is given in the review by Pandey et al. [6].

The Localized Surface Plasmon Resonance (LSPR) technique uses the SPR signal, which appears on metal (gold, silver) nanoparticles immobilized on a dielectric surface (usually on an optical fiber). The light beam is provided by the fiber and, after passing through the sensing part, is directed to a spectrometer. The red shift of the spectrum caused by the presence of an adsorbed analyte on the biosensor is the analytical signal. An advantage of LSPR is the possibility of miniaturizing the instrument and biosensor. More information about LSPR-based fiber optic analyte sensors is given in the review by Chauhan and Singh [7].

The array SPRi technique uses the Kretschmann configuration, with the biosensor containing a gold chip fixed to a prism with immersion oil. The chip surface contains an array of measuring points separated by a polymer. The array is divided into nine measuring cells separated by a hydrophobic paint. Each cell contains a dozen of measuring points. The chip architecture makes it possible to measure nine separate samples simultaneously. The SPRi signal is measured twice: before interaction and after interaction with the analyte. These measurements are performed after the removal of processing solutions, that is, in a dry state.

Most SPR, LSPR and SPRi biosensors contain a suitable antibody as a sensing element, attached to a gold surface via a bifunctional linker containing a thiol group and alternatively carboxyl or amino groups. In array SPRi, aminothiol is usually used as the linker, while in SPR biosensors, the linker is often carboxythiol, or alternatively, carboxylated dextran. In both cases, the EDC/NHS protocol is used for the formation of amide bonds between the antibody and the linker. The use of an aminothiol linker seems to provide the advantage of a perpendicular orientation of the attached antibody relative to the gold chip surface because the Fc region of the antibody contains a majority of carboxyl groups. In the case of the fluidic SPR technique, a simple antibody-containing biosensor suffers from a lack of sensitivity for the determination of molecular biomarkers. Therefore, a more complex biosensor is used to enhance the SPR signal; a sandwich structure with a secondary antibody conjugated with gold nanoparticles is used [2,3,4,5]. Apart from the antibody, a suitable inhibitor or aptamer may be used as the sensing element of the biosensor.

Unlike the fluidic SPR technique, array SPRi has great potential in the determination of molecular biomarkers. These two techniques differ in terms of the presence of aqueous solution during SPR measurement: in the case of array SPRi, an aqueous solution is removed before measurement. In addition, the biosensor is formed in situ during measurement in the case of fluidic SPR, while in array SPR, the biosensor is formed ex situ before measurement. These two differences enable biomarker determination without any signal enhancement or preliminary preconcentration in the case of array SPRi.

The array SPRi technique offers new opportunities for the determination of biomarkers. The technique is used for the determination of molecular biomarkers in body fluids such as blood serum/plasma, urine, saliva, etc., in what is called ‘liquid biopsy’. Almost 30 biosensors operating with this technique have been developed, including some for the determination of the cancer markers CA 125 [8], HE4 [9] and CEA [10], as well as the physiologically significant laminin 5 [11], collagen type IV [12], fibronectin [13], leptin [14], MMP 1 [15], MMP 2 [16], and many others. Numerous biosensors have been used in clinical investigations, such as the detection of bladder cancer by measuring serous podoplanin [17] or aromatase [18], or determination of progress in the healing of thermal injuries based on measurements of serous UCHL 1 [19], MMP-2 [20] or 20S proteasome [21].

Interleukin-6 (IL-6) plays an important role in the progression of many diseases, including COVID-19. IL-6 plays a critical role in the control of inflammation and is essential in cases of autoimmune inflammation such as multiple sclerosis, rheumatoid arthritis, and systemic lupus erythematous [22]. It also plays a fundamental role in the advanced stage of COVID-19, where it participates in the formation of a dangerous cytokine storm, which is reflected by high IL-6 concentration in blood serum. Elevated IL-6 levels are characteristic of ovarian cancer [23], gastric cancer [24,25] and esophageal cancer [26]. IL-6 is a glycoprotein consisting of 212 amino acids with MW in the range 21–30 kDa [27]. The plasma/serum concentration median of healthy controls varies from 1.2 [28] to 3.96 pg/mL [23] or 3.2–5.0 pg/mL [29], while the highest concentration is reported as 19.9 pg/mL. Much higher serous IL-6 concentrations have been reported from patients with ovarian cancer (up to 41.23 pg/mL; [23]) or acute appendicitis (up to above 1000 pg/mL; [29]). IL-6 is determined using the ELISA immunoassay, the Luminex 200 compact analyzer system [24] or the multiplex analyte profiling technology xMAP [30]. Diagnostics Roche offers the COBAS E-411 analyzer, which determines IL-6 using the electrochemiluminescence (ECL) method. All these methods use labels. The determination of IL-6 by fluidic SPR has also been reported [31]. A sandwich biosensor composed of two antibodies enables the determination of IL-6 at levels of ng/mL, which is insufficient for the determination of markers in blood serum or plasma. Array SPRi is a label-free technique.

The aim of this study was to develop a biosensor for IL-6 determination in human blood plasma to determine its analytical characteristics and to provide examples of its clinical application. Two versions of the biosensor were considered: the first using a suitable antibody as a receptor and the second using an IL-6 inhibitor as a receptor. Initial experiments showed that mouse monoclonal anti-IL-6 antibody and galiellalactone specific to IL-6 are well suited to this purpose. Cysteamine has been found to be a suitable linker for the antibody, with the formation of covalent bonds with carboxylic groups of the antibody by the EDC/NHS protocol. The optimum concentration of the mouse monoclonal anti-IL-6 antibody used as a receptor was established at 60 pg mL^−1^. The inhibitor galiellalactone was immobilized via an octadecanethiol (ODM) linker, using hydrophobic interactions between the inhibitor and linker. Because galiellalactone does not contain carboxyl or amino groups in its structure, its immobilization via EDC/NHS protocol is impossible. The optimum concentration of galiellalactone as a receptor was established at 50 pg mL^−1^. Both biosensors were tested in parallel.

## 2. Methods

### 2.1. Materials

#### 2.1.1. Reagents

Recombinant Human IL-6 (Abcam plc, Cambridge, MA, USA), mouse monoclonal anti- IL-6 antibody (Abcam plc, Cambridge, MA, USA), inhibitor galiellalactone specific of IL-6 (Santa Cruz Biotechnology, Dallas, TX, USA), albumin, leptin, metalloproteinase-2, recombinant human CEA, recombinant human CA125-MUC16, cysteamine hydrochloride, N-ethyl-N′-(3-dimethylaminopropyl) carbodiimide (EDC) (Sigma Steinheim, Steinheim, Germany), N-Hydroxysuccinimide (NHS) (Aldrich, Munich, Germany), octadecanethiol (ODM) were used. HBS-ES solution pH = 7.4 (0.01 M HEPES, 0.15 M sodium chloride, 0.005% Tween 20, 3 mM EDTA), photopolymer ELPEMER SD 2054, hydrophobic protective paint SD 2368 UV SG-DG (Peters, Kempen, Germany, www.peters.de/, accessed on 2 June 2022), Phosphate Buffered Saline (PBS) pH = 7.4, carbonate buffer pH = 8.5 (BIOMED, Lublin, Poland) were used as received. Aqueous solutions were prepared with MilliQ water (Simplicity^®^MILLIPORE, Merck KGaA, Darmstadt, Germany).

#### 2.1.2. Biological Samples

Blood samples were taken from a vein in the arm. Plasma was prepared according to standard procedures. Plasma samples were immediately frozen and stored at −80 °C. Consent for this study was obtained from the Bioethics Committee of the Medical University of Białystok (Białystok, Poland), and written informed consent was obtained from all patients.

### 2.2. Biosensor Preparation

#### 2.2.1. Chip Preparation

Gold chips were manufactured as described in previous papers [11,32]. The gold surface of the chip was covered with a photopolymer and hydrophobic paint, a procedure described previously. As a result, 9 × 12 free gold surfaces were obtained. With the use of this chip, nine different solutions can be simultaneously measured without mixing the tested solutions, and twelve single SPRi measurements can be performed from one solution.

#### 2.2.2. Antibody Immobilization

The chips were rinsed with ethanol and water and dried under a stream of argon. They were then immersed in 20 mM ethanolic cysteamine for at least 12 h, then rinsed again with ethanol and water and dried under a stream of argon. The next step was to immobilize the mouse monoclonal anti-IL-6 antibody on the surface of the biosensor (biosensor (a). The antibody solution in PBS buffer (60 pg mL^−1^) was activated with NHS (50 mM) and EDC (200 mM) in a carbonate buffer (pH = 8.5). The activated antibody was then placed on a cysteamine-modified surface and incubated at 37 °C for 1 h. A workflow for the formation of a biosensor (a) with mouse monoclonal anti-IL 6-antibody as the receptor is shown in Figure 1.

#### 2.2.3. Immobilization of Galiellalactone

The biosensor was immersed in 1-octadecanethiol (ODM) in ethanol (20 nM) for 24 h at room temperature. Then the biosensor (biosensor (i)) was washed several times in ethanol and water and dried under a stream of argon. The chip prepared in this way can be stored and used for subsequent measurements. The specific inhibitor galiellalactone (50 pg mL^−1^) was then applied to the active sites of the biosensor (i) and incubated for 24 h at room temperature. After this time, the process of immobilization of the inhibitor was completed. A biosensor (i) prepared in this way should be used directly for analysis. A workflow of the formation of a biosensor (i) with galiellalactone as the receptor is shown in Figure 2.

### 2.3. SPRi Measurements

SPRi measurements were performed as previously described, using a homemade SPRi apparatus [11,32]. The signal was measured twice from the recorded images, after the receptor (antibody or inhibitor) was immobilized and then after interaction with the IL-6 solution. IL-6 solutions or diluted plasma samples were placed directly on the prepared biosensor for 10 min to allow interaction with the receptor (antibody or inhibitor). The volume of the sample applied to each measuring field was 3 μL. After this time, the biosensor was washed with water to remove unbound particles from the surface. SPRi measurements were made at a constant light angle. Non-specific binding was monitored by measuring the SPRi signal at a site on the chip. Non-specific binding was minimized by preparing the samples in PBS buffer and placing BSA in PBS buffer on the chip. The SPRi signal was obtained as the difference between the signals before and after the interaction with the analyzed sample for each active site separately. The results were evaluated using the calibration curve.

### 2.4. Standard Method Used for Validation of the Developed Biosensor

The standard method used to determine the IL-6 marker was the Elecsys IL-6 electrochemiluminescence test using the COBAS E-411 analyzer. The test was performed by a routine diagnostic laboratory.

## 3. Results

### 3.1. Calibration Curves for Both Biosensors

Calibration experiments were performed for both biosensors within the IL-6 concentration range 1–100 pg mL^−1^. Experiments were performed as described in Section 2.2.2 and Section 2.3 for the biosensor containing the antibody and in Section 2.2.3 and Section 2.3 for the biosensor containing the inhibitor galiellalactone. The results are shown in Figure 3. Both curves exhibit linearity up to 20 pg mL^−1^ of IL-6, with a slightly higher slope for the biosensor with the antibody. This range is well matched to the levels of IL-6 found in serous samples.

### 3.2. Selectivity of the Biosensor with Mouse Monoclonal Anti-IL-6 Antibody as Receptor

The selectivity of a biosensor is a crucial factor for its application in determining an analyte in real samples. However, the choice of potential interferents is often difficult. All of the body fluids used in diagnostics contain very many potential interferents. Therefore, the potential interferents chosen in this investigation are considered as examples rather than specific knowledge-based choices. Albumin, leptin, CEA, CA-125 and metalloprotein-2 were selected as interfering substances. Albumin is always present in human plasma/serum in excess. The other substances are frequently determined in blood plasma/serum in parallel with the determination of IL-6. The experiments were performed at a constant IL-6 spike (20 pg mL^−1^) and 1000 times excess of the interferent. The results are shown in Table 1.

The results show that none of the tested interferents has a significant influence on the results. The recoveries are 100 ± 3%.

### 3.3. Selectivity of the Biosensor with the Inhibitor Galiellalactone as Receptor

The same potential interferents were used in this investigation. The experiments were performed at a constant IL-6 spike (20 pg mL^−1^) and 1000 times excess of the interferent. The results are shown in Table 2.

None of the tested interferents has a significant influence on the results. The recoveries are 100 ± 4%.

### 3.4. Analytical Characteristics of the Developed Biosensors

Analytical characteristics—precision, recovery, and limits of detection (LOD) and quantification (LOQ)—were determined for both developed biosensors under model conditions. Three series of measurements were performed with spikes of IL-6 equal to 3.10, 10 and 20 pg/mL. The results are shown in Table 3.

The precision and recoveries of the developed biosensors are typical for this type of determination and are satisfactory. The recovery of IL-6 concentrations was found to be in the range 97.4–107% for the biosensor with the antibody as the receptor and 101–105% for the biosensor with the inhibitor as the receptor. The limit of detection, calculated from the standard deviation (3SD), is 0.90 pg mL^−1^ for the biosensor with the antibody as the receptor and 0.33 pg mL^−1^ for the biosensor with the inhibitor as the receptor. The SD values of the lowest spike were used in this calculation. Thus, the limits of quantification (10 SD) are 3.0 and 1.10 pg mL^−1^, respectively.

### 3.5. Determination of IL-6 Concentration in Biological Samples by the Two Biosensors

The determination of IL-6 in blood plasma samples is a very significant test for both developed biosensors because plasma contains many known and unknown potential interferents. These experiments were performed with 10 different samples of blood plasma assumed to have divergent IL-6 concentrations. IL-6 was determined in each sample by means of the two biosensors. Most of the plasma samples were diluted ten times with PBS buffer before measurement. The results are compared in Figure 4.

The comparison shows that the results obtained using the biosensor with the antibody as the receptor and using the biosensor with the inhibitor as the receptor are equivalent (R^2^ = 0.97 and slope close to 1). It is worth noting that the structures of the two biosensors are entirely different.

### 3.6. Validation of the Developed Biosensor with Mouse Monoclonal Anti-IL-6 Antibody as Receptor by Comparison with a Standard Method

Il-6 was determined in a series of 12 different blood plasma samples assumed to contain different biomarker concentrations, using the biosensor with the antibody as the receptor and by means of the electrochemiluminescence test performed on a COBAS E-411 analyzer in a routine diagnostic laboratory. The comparison of the results is shown in Figure 5. The slope close to 1 and R^2^ equal to 0.9989 indicate the high equivalence of the results. Thus, the biosensor with the antibody as the receptor has been validated. Indirectly, the biosensor with the inhibitor galiellalactone has also been validated, given the equivalence of the two biosensors, as confirmed by the results in Figure 4.

### 3.7. Example of Clinical Application of the Developed Biosensors—Resection of Ovarian Tumor and Endometrial Cyst

To exhibit the potential of the two developed biosensors in clinical investigation, a series of measurements of IL-6 concentration in blood plasma samples taken during the resection of an ovarian tumor and endometrial cyst was performed. The samples were taken before resection and 6 h, 24 h and 5 days after resection of the ovarian tumor and 5 and 24 h after resection of the endometrial cyst. A control sample from a healthy subject was included. The measurements were performed using both of the developed biosensors. The results are shown in Figure 6 and Figure 7.

It is clear that both biosensors are suitable tools for such observations, showing the gradual decrease in IL-6 concentration after resections. Each of the two biosensors produces the same picture of changes in IL-6 concentration during the treatment.

## 4. Discussion

Due to the lower level of IL-6 concentration, the determination of IL-6 in blood plasma/serum samples is a more difficult task than in the case of the majority of biomarkers. Most biomarkers are present in serous samples at the ng mL^−1^ level, while IL-6 occurs at the pg mL^−1^ level. In spite of this obstacle, the developed biosensors are able to determine IL-6 at the required level of concentration without any signal enhancement or preliminary preconcentration. This highlights the great potential of array SPRi in the determination of molecular biomarkers and in the development of ‘liquid biopsy’. This work has added two new array SPRi biosensors to the set of those already in use. The advantage of the array SPRi biosensors in the determination of IL-6 is obvious when they are compared with the SPR biosensor developed previously [31]. A simple biosensor, as described by Chou et al. [31], consisting of an antibody and a linker, does not react to an increase in IL-6 concentration.

The sandwich biosensor (antibody–analyte–secondary antibody) developed by those authors enables the determination of IL-6 at the level of a few ng mL^−1^, while the biosensors described in this paper enable the determination of IL-6 at the level of pg mL^−1^, which is necessary for the determination of the biomarker in blood plasma/serum samples.

Despite the fact that the IL-6 concentration is determined at the pg mL^−1^ level, the precision of IL-6 determination is satisfactory for both developed biosensors, as are the recoveries. The precision does not exceed 10%, and the recoveries are between 97.4% and 107% for the biosensor with the antibody as the receptor and between 101% and 105% in the case of the biosensor with the inhibitor as the receptor. The linear response range is between 3 (LOQ) and 20 pg mL^−1^ in the case of the biosensor with the antibody as the receptor and between 1.1 (LOQ) and 20 pg mL^−1^ in the case of the biosensor with the inhibitor as the receptor. In the case of samples with IL-6 concentrations greater than 20 pg mL^−1^, appropriate dilution with PBS buffer was used.

Both biosensors have been validated by the determination of IL-6 in real blood plasma samples simultaneously using the developed biosensor with the antibody as the receptor and by the electrochemiluminescence test performed on a COBAS E-411 analyzer, the tool used in diagnostic laboratories, with excellent agreement of the results (see Figure 5). The other biosensor with the inhibitor as the receptor has been validated indirectly because the results obtained with this biosensor are in good agreement with the results obtained with the biosensor with the antibody as the receptor (see Figure 3). This consistency of the results obtained by both biosensors is also visible in the series of results for IL-6 concentration in the samples of blood plasma from patients undergoing ovarian tumor resection (see Figure 6) and endometrial cyst resection (see Figure 7).

Both developed biosensors exhibit good selectivity, as is evidenced in a series of experiments with potential interferents: albumin, CEA, CA 125, leptin and MMP2. The results are shown in Table 1 for the biosensor with mouse monoclonal anti- IL-6 antibody as a receptor and in Table 2 for the biosensor with the galiellalactone receptor. Apart from albumin, which is present in human blood in excess, the selection of the other interferents was somewhat random. The recoveries do not exceed 100 ± 4% for either biosensor. However, selectivity is best examined by means of experiments with blood plasma, which contains all potential interferents. The fact that two entirely different biosensors and the electrochemiluminescence test used as the reference method exhibit basically the same results of IL-6 in plasma samples supports the hypothesis of the lack of interference.

Considering the good agreement of the results obtained using both of the developed biosensors, it is worth noting that the structures of these biosensors are entirely different. The biosensor with the antibody as the receptor consists of the mouse monoclonal anti-IL-6 antibody covalently attached to the gold surface via a cysteamine linker. Amide bonds are formed between the linker and the antibody via the NHS/EDS protocol. The application of the cysteamine linker ensures the perpendicular orientation of the antibody with respect to the chip surface. This is because the Fc region of the antibody contains numerous carboxyl groups capable of reacting with the amine groups of cysteamine. The antibody’s reactive part binding the antigen is located opposite the Fc domain. The biosensor with the inhibitor as the receptor consists of the inhibitor galiellalactone attached to the gold surface via 1-octadecanethiol (ODM). The inhibitor is bonded to the ODM linker by hydrophobic interaction. The structure of galiellalactone shows that the compound has no functional groups suitable for covalent bond formation while indicating its hydrophobicity. The ODM linker is fixed on the gold surface via its thiol group. The long aliphatic chain ensures a strong hydrophobic interaction with galiellalactone. It is worth noting that both developed biosensors have a very simple structure, as is characteristic for all array SPRi biosensors. Despite the entirely different structures, both biosensors give very similar results for IL-6 determination. This agreement provides additional validation of both biosensors because it is unlikely that two erroneous biosensors would give equally erroneous results.

Each of the developed biosensors can be successfully used in monitoring changes in IL-6 concentration in blood plasma/serum after surgery and can thus be successful tools in liquid biopsy. Only 3 µL of plasma or serum is needed for the determination of IL-6. Nine different samples can be measured simultaneously.

## 5. Conclusions

1. An array SPRi biosensor for the determination of interleukin-6 has been developed, consisting of a cysteamine linker and the mouse monoclonal anti-IL-6 antibody. The biosensor is highly selective and exhibits good precision and recovery. The biosensor has a linear response range between 3.0 (LOQ) and 20 pg mL^−1^, which is well-suited for the IL-6 determination in human plasma. The biosensor was validated by parallel plasma Il-6 determination using the electrochemiluminescence test, with a good agreement of the results.

2. A second array SPRi biosensor for the determination of IL-6 has been developed. The biosensor has an entirely different structure from the previous biosensor and contains 1-octadecanethiol as the linker and galiellalactone as the receptor. The biosensor is highly selective and exhibits good precision and recovery. It has a linear response range between 1.1 (LOQ) and 20 pg mL^−1^, which is well-suited for IL-6 determination in human plasma. The biosensor was validated by parallel plasma Il-6 determination using the array SPRi biosensor with mouse monoclonal anti-IL-6 antibody as the sensing element with a good agreement of the results.

3. The applicability of both biosensors was verified in clinical investigations of ovarian cancer and endometrial cyst resection by the measurement of IL-6 levels in the plasma of patients, showing agreement of the results obtained with both biosensors.

## Figures and Tables

**Figure 1 biosensors-12-00412-f001:**
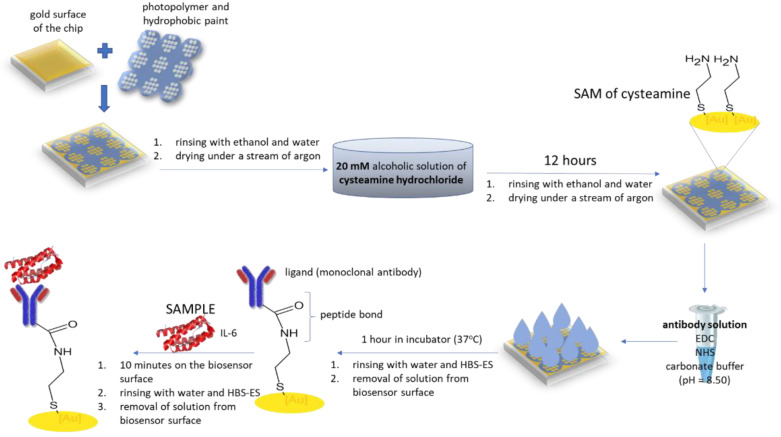
Workflow of the formation of a biosensor (a) with mouse monoclonal anti-IL 6-antibody as the receptor.

**Figure 2 biosensors-12-00412-f002:**
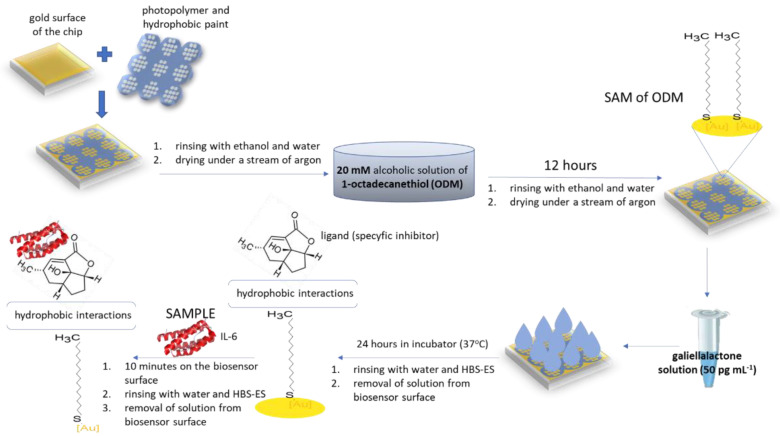
Workflow of the formation of a biosensor (i) with galiellalactone as the receptor.

**Figure 3 biosensors-12-00412-f003:**
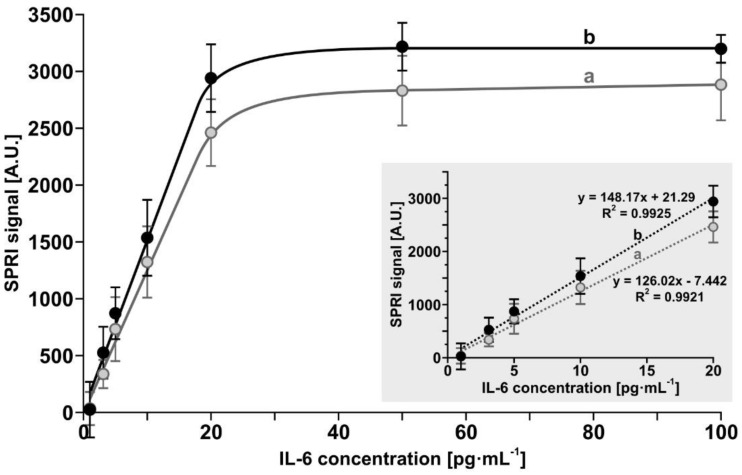
Dependence of the SPRi signal (arbitrary units) on the concentration of IL-6 for the biosensors containing: (a) mouse monoclonal anti-IL-6 antibody, (b) galiellalactone. Error bars (confidence limits) were calculated for 12 independent measurements for each concentration at the 95% confidence level. The inset shows the linearity range.

**Figure 4 biosensors-12-00412-f004:**
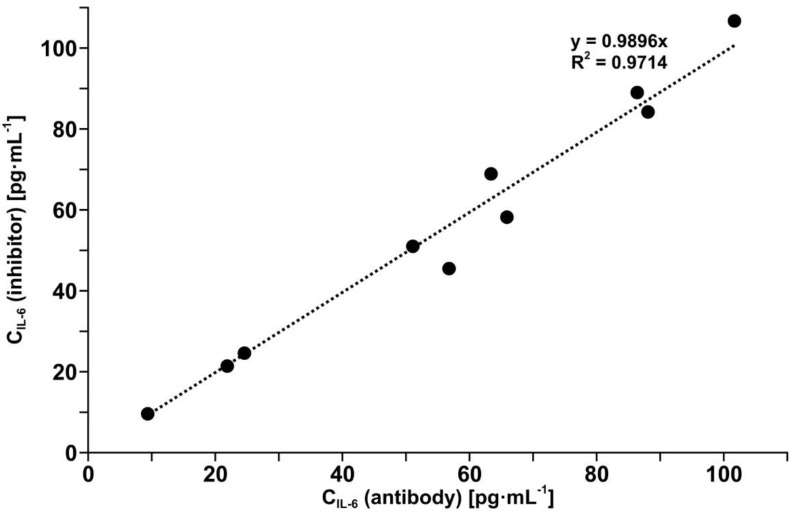
Comparison of results of IL-6 determination in blood plasma as determined by the biosensor with mouse monoclonal anti-IL-6 antibody and the biosensor with the inhibitor galiellalactone.

**Figure 5 biosensors-12-00412-f005:**
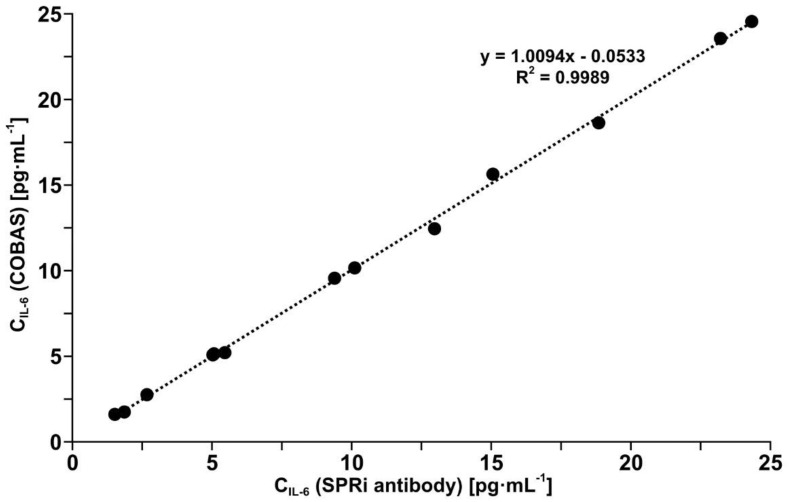
Comparison of the results of IL-6 in blood plasma determination using the biosensor with the mouse monoclonal anti-IL-6 antibody and by the electrochemiluminescence test performed on a COBAS E-411 analyzer.

**Figure 6 biosensors-12-00412-f006:**
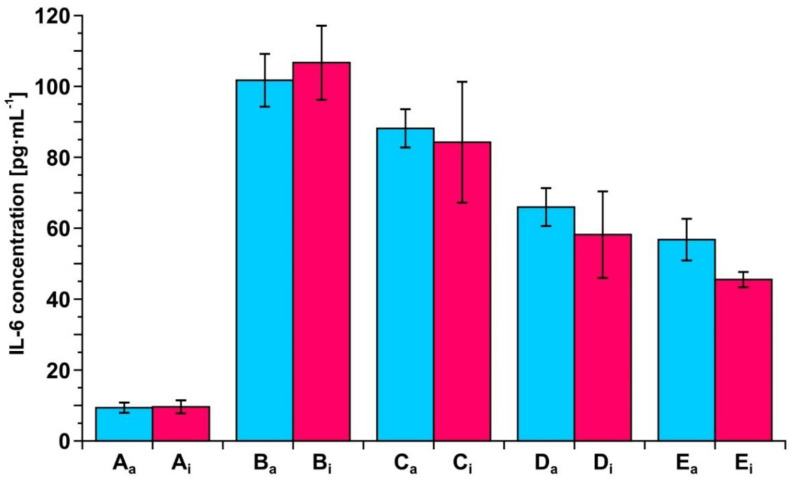
Changes in IL-6 concentration in the blood plasma of a patient before and after resection of ovarian tumor, as determined by the biosensor with the antibody as the receptor (blue) or the biosensor with the inhibitor as the receptor (red). A: control, B: before resection, C: 6 h, D: 24 h and E: 5 days after resection.

**Figure 7 biosensors-12-00412-f007:**
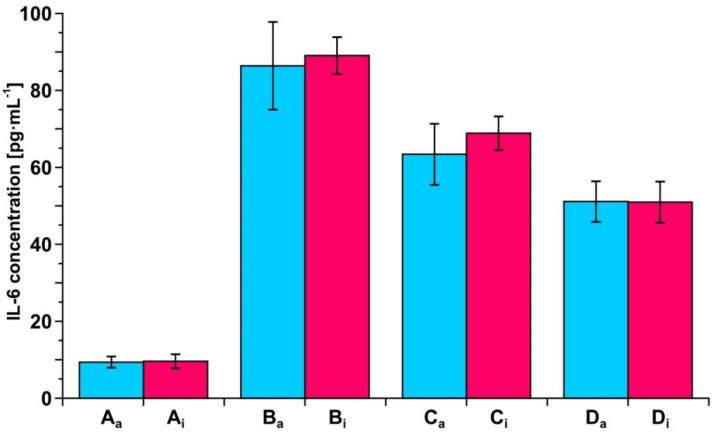
Changes in IL-6 concentration in the blood plasma of a patient before and after resection of endometrial cyst, as determined by the biosensor with the antibody as the receptor (blue) or the biosensor with the inhibitor as the receptor (red). A: control, B: before resection, C: 5 h, and D: 24 h hours after resection.

**Table 1 biosensors-12-00412-t001:** Influence of excess of human albumin, leptin, CEA, CA-125 and metalloproteinase-2 (1000:1) on the results of determination of IL-6 concentration by the biosensor with antibody as the receptor (biosensor (a)).

Interferent	IL-6 Spike (pg mL^−1^)	Found IL-6 (pg mL^−1^)	Recovery (%)
CEA	20.0	19.4 ± 2.68	97.0
CA 125	20.0	19.7 ± 1.31	98.5
Leptin	20.0	20.6 ± 3.05	103
MMP 2	20.0	20.5 ± 3.94	102
Albumin	20.0	19.0 ± 5.88	97.9

**Table 2 biosensors-12-00412-t002:** Influence of excess of human albumin, leptin, CEA, CA-125 and metalloproteinase-2 (1000:1) on the results of determination of IL-6 concentration by the biosensor with the inhibitor as the receptor (biosensor (i)).

Interferent	IL-6 Spike (pg mL^−1^)	Found IL-6 (pg mL^−1^)	Recovery (%)
CEA	20.0	20.8 ± 1.34	104
CA 125	20.0	19.7 ± 9.2	98.6
Leptin	20.0	19.9 ± 1.07	99.4
MMP 2	20.0	20.5 ± 8.75	102
albumin	20.0	20.5 ± 1.64	102

**Table 3 biosensors-12-00412-t003:** Precision and recovery of the determination of IL-6 spikes by biosensor (a) and biosensor (i).

Biosensor with	IL-6 Spike (pg mL^−1^)	Found (pg mL^−1^)	SD (pg mL^−1^)	RSD (%)	Recovery (%)
antibody	3.1	3.02	0.30	9.7	97.4
	10	10.7	0.98	9.8	107
	20	19.6	1.6	8.0	97.9
inhibitor	3.1	3.24	0.11	3.5	104
	10	10.5	0.93	9.3	105
	20	20.2	1.8	8.9	101

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
