# Peer review of "Two Biosensors for the Determination of Interleukin-6 in Blood Plasma by Array SPRi"

_biosensors, 2022, doi:10.3390/bios12060412_

Round 1

Author Response

We thank the Reviewer for the generally positive evaluation of the manuscript and for the critical remarks.

  1. Some of the sentences are too short. For example:
  • Both biosensors are specific for IL-6.
  • Both biosensors were validated.

These two sentences are used in the Abstract, where the information given is synthetic and the number of words is limited

  1. In the Keywords, the last semicolon (;) should be changed to dot (.).

Recommended change has been introduced.

  1. Some sentences are unclear or not logical. For example, ‘Body fluids such as blood plasma or serum, urine, saliva, cerebrospinal fluid, etc contain a vast amount of potentially useful diagnostic information which is so far unavailable due to the lack methods for its discovery.’ Some body fluids are easily available. The sentence can be written in a way that reflects the intention of the authors.

This sentence has been rewritten (lines 32-34).

  1. The integration of the introduction is not satisfying. The linking between the sentences are not good.

We are grateful  for this comment.

  1. Some sentences are not appropriate for formal writing. For example:
  • doctors’ expectations are much higher …

The term doctors’ has been replaced by oncologists’(line 44).

  1. There are some extra spacing between words in some parts for example lines 54 and 92.

Extra spacing have been removed (lines 54 and 92)

  1. Although the applied chip and the optical setup have been previously used in previous publication, it would be good if the author add a figure showing the chip and the applied optical system. In addition, figures showing the receptors and linkers seems essential to be inserted in the manuscript.

Two figures have been added showing the chip, receptors and linkers. (Fig. 1 and 2)

  1. In Reviewer’s opinion, there are not two different biosensors. There is one biosensor where different receptors and linkers have been applied.

We disagree with this opinion.

  1. The meaning and the goal for ‘Added IL-6’ and ‘Added IL-6’ are not clear from the text.

The term ‘added IL 6’ has been replaced by ‘spike of IL 6’ in Tables 1-3.

  1. The title of 3.7 should be revised:

3.7. Example of clinical application of the developed biosensors: Resection of ovarian tumor

The title of 3.7 has been supplemented  (331-332)

  1. The opening sentence of the Discussion needs to be rephrased.

This sentence has been rewritten. (lines 365-367)

  1. In the discussion, the authors have compared their biosensors with the ones published previously. A comparison table will be helpful in this case.

Only one SPR  biosensor has been reported in the literature.

  1. What is the sensitivity of the proposed biosensor?

The limit of detection, calculated from the standard deviation (3SD), is 0.90 pg mL-1 for the biosensor with the antibody as the receptor and 0.33 pg mL-1 for the biosensor with the inhibitor as the receptor. (lines 300-302)

  1. The conclusion has not been written in the regular format.

We used the MDPI template

Reviewer 2 Report

In this work, the authors presented two specific biosensors for the determination of IL-6 in blood plasma by array SPRi. Authors have used two receptors, the mouse mono- 18 clonal anti-IL-6 antibody and galiellalactone, for IL-6 recognition for each biosensor. The work could be interesting for the readers and consider the good literature review and results. Nonetheless, some important aspects for measuring principle and biosensor performance are missing. Therefore, I suggest a major revision before recommending this manuscript for publication.

1. In the introduction part, more references should be added related to theory and the current research state of SPR.

2. In line 59, I think the sentence ‘Fluidic SPR is the most popular SPR technique’ is overexpression. Again, relative references are required.

3. In line 135, I think you have proposed two biosensors in this work, not a biosensor.

4. In line 136, the its ambiguous.

5. In section 2.2.1, the chip structures and detect principles should be described.

6. In section 2.2.2, why did you choose a buffer with PH=8.5 in your experiment?

7. The descriptions in sections 2.2.2 and 2.2.3 are illogical. They are related to two different sensors, I suggest the authors clarify the sensors as numbers, such as sensor 1 and sensor 2.

8. In the measurement for IL-6, how to clear the bonding IL-6 on the sensors after each measurement?

9. What does the recovery mean in the Tables?

10. Nothing is well expressed about the reproducibility of the sensors by testing several identical probes.

11. How about the reuse of the sensor probe?  

12. The LOD calculation should be explained.

Author Response

The authors express thanks for the Reviewer’s generally positive evaluation of the paper and for the stimulating comments which have enabled the paper to be improved.

1.In the introduction part, more references should be added related to theory and the current research state of SPR.

Two references have been added with appropriate comments (lines 72-74  and 81-82) and references 6 and 7

2.In line 59, I think the sentence ‘Fluidic SPR is the most popular SPR technique’ is overexpression. Again, relative references are required.

This sentence has been modified  and references have been added (line 59)

3.In line 135, I think you have proposed two biosensors in this work, not ‘a biosensor’.

4.In line 136, the ‘its’ ambiguous.

Two biosensors are the final result; however, our original task was to develop minimum of one biosensor.

  1. In section 2.2.1, the chip structures and detect principles should be described.

Two figures containing work flows have been added (Fig.1 and 2)

  1. In section 2.2.2, why did you choose a buffer with PH=8.5 in your experiment?

A pH value of  8.5 is optimum for the EDC/NHS protocol

  1. The descriptions in sections 2.2.2 and 2.2.3 are illogical. They are related to two different sensors, I suggest the authors clarify the sensors as numbers, such as sensor 1 and sensor 2.

This suggestion has been adopted, and the biosensors are labelled  as biosensor (a) and biosensor (i)

  1. In the measurement for IL-6, how to clear the bonding IL-6 on the sensors after each measurement?

The only regeneration of used biosensors was the removal of IL-6, receptor and linker by treatment with a mixture of sodium hydroxide and Triton – X 100.

  1. What does the ‘recovery’ mean in the Tables?
  2. Nothing is well expressed about the reproducibility of the sensors by testing several identical probes.

The recovery means the percentage of the introduced spike that was determined. This is a typical procedure in analytical chemistry, and represents accuracy of measurement.

  1. How about the reuse of the sensor probe?

 The only regeneration of used biosensors was the removal of IL-6, receptor and linker by treatment with a mixture of sodium hydroxide and Triton – X 100.

  1. The LOD calculation should be explained.

The LOD calculation is explained in lines 298-300. The SD of the lowest spike was used in this calculation

Reviewer 3 Report

In this manuscript, the authors describe two SPRi based biosensors for quantification of interlekin-6 in blood samples. The authors compared the two biosensors and other existing methods to demonstrate that both biosensors have decent limit of detection/quantification, precision, and recoveries. The results are overall solid, while the manuscript does not clearly specify the advantages of these new biosensors over existing ones.

Major points:

1.     I suggest adding a schematic/cartoon figure to illustrate the work flow with the two biosensors, such as the process of sample loading/analyte binding/washing. This can help readers to easily understand the working mechanism of the biosensors, especially the inhibitor one.

2.     The introduction section can be improved. First, it is a little redundant and can be more concise, especially the explanation of different techniques of SPR. Second, the motivation of developing new biosensors for IL-6 is not clear. Is there any limitation of previous methods that new biosensor can overcome?

3.     In figure 2, the authors quantified samples with high concentration of IL-6 (up to 100 pg/mL). It is confusing here because in Figure 1, the signals are saturated after 20 pg/mL and readers may wonder how the 100 pg/mL was quantified. In line 380, the authors mentioned using dilution, but I think it would be better to describe it in the experimental section and maybe also in section 3.5. 

I also think the quantification method is not mentioned in the manuscript. Did the authors use calibration curve, standard addition, or other methods?

Minor points:

1.     Line 90: give the full name of NHS/EDS. Isn’t it EDC?

2.     Line 127: “reported ‘from’ patients with …”

3.     Line 233: “calculated ‘based on’ 12 independent measurements”. Also, please indicate whether error bars indicate standard deviation, coefficient of variation or something else.

4.     Figure 1: the inset should be briefly described in the caption.

5.     Line 247: “The recoveries are 100 ± 3%”.

6.     The lines in Table 3 are strange. Please put the table in a single page.

7.     Line 280, it will be better to discuss why the biosensor using the inhibitor can give a better limit of detection.

8.     Line 401: “supports the ‘hypothesis’ of the lack of interference”.

9.     In conclusions section, paragraph indentation is not consistent.

Author Response

We thank the Reviewer for the generally positive evaluation of the manuscript and for the critical remarks which have enabled the paper to be improved.

  1. I suggest adding a schematic/cartoon figure to illustrate the work flow with the two biosensors, such as the process of sample loading/analyte binding/washing. This can help readers to easily understand the working mechanism of the biosensors, especially the inhibitor one.

The suggested work flows  have been introduced (Figs.1 and 2)

  1. The introduction section can be improved. First, it is a little redundant and can be more concise, especially the explanation of different techniques of SPR. Second, the motivation of developing new biosensors for IL-6 is not clear. Is there any limitation of previous methods that new biosensor can overcome?

All applied  methods use labels. Array SPRi is lablel-free and  sufficiently sensitive. (lines 137-141). Several sentences of the Introduction was removed.

  1. In figure 2, the authors quantified samples with high concentration of IL-6 (up to 100 pg/mL). It is confusing here because in Figure 1, the signals are saturated after 20 pg/mL and readers may wonder how the 100 pg/mL was quantified. In line 380, the authors mentioned using dilution, but I think it would be better to describe it in the experimental section and maybe also in section 3.5.

Most of the plasma samples was ten-times diluted with PBS buffer before measurement (lines 311-312)

I also think the quantification method is not mentioned in the manuscript. Did the authors use calibration curve, standard addition, or other methods?

The results were evaluated using the calibration curve. (lines 231-232)

Minor points:

Line 90: give the full name of NHS/EDS. Isn’t it EDC?

This term has been corrected (line 95)

Line 127: “reported ‘from’ patients with …”

This sentence has been corrected (lines 132-134)

Line 233: “calculated ‘based on’ 12 independent measurements”. Also, please indicate whether error bars indicate standard deviation, coefficient of variation or something else.

Errors bars were calculated on the basis confidence limits . This information has been inserted in the caption of Fig. 3

Figure 1: the inset should be briefly described in the caption

This suggestion has been introduced (line 254)

 Line 247: “The recoveries are 100 ± 3%”

The suggested correction has been made  (line 268).

            The lines in Table 3 are strange. Please put the table in a single page.

The lines in Table 5 (new number) have been modified.

Line 280, it will be better to discuss why the biosensor using the inhibitor can give a better limit of detection.

We have no explanation for this difference between the biosensors

Line 401: “supports the ‘hypothesis’ of the lack of interference”

This sentence has been modified following the suggestion (lines 425-427)

In conclusions section, paragraph indentation is not consistent.

The suggested change has been made (line 460)

Round 2

Reviewer 1 Report

Dear Authors, 

The responses to the Reviewer are too short. They can be followed up by more concise explanation. However, the addition of the figures improved the manuscript in a good way so I recommend it for acceptance.

Reviewer 2 Report

I think the manuscript is suitable for publication now.

Reviewer 3 Report

The authors addressed all my comments.